materials science/electromagnetism

room temperature ferromagnetism, perovskite niobates, BaNbO$_3$

**Authors for correspondence:**
Fei Zhou
e-mail: angel.flyfly@hotmail.com
Jingchuan Zhu
e-mail: fgms@hit.edu.cn

†These authors contribute equally to this study. This article has been edited by the Royal Society of Chemistry, including the commissioning, peer review process and editorial aspects up to the point of acceptance.

# Increased room temperature ferromagnetism in Co-doped tetrahedral perovskite niobates

Yi Zhou[1,†], Qing He[1,†], Fei Zhou[1,2], Xingqi Liao[1], Yong Liu[1,3], Zhonghong Lai[4], Mingqing Liao[1], Tianyi Han[1], Yudong Huang[2] and Jingchuan Zhu[1,3,5]

[1]School of Materials Science and Engineering, [2]MIIT Key Laboratory of Critical Materials Technology for New Energy Conversion and Storage, School of Chemistry and Chemical Engineering, [3]National Key Laboratory for Precision Hot Processing of Metals, [4]Center of Analysis and Measurement, and [5]National Key Laboratory of Science and Technology on Advanced Composites in Special Environments, Harbin Institute of Technology, 150001 Harbin, People's Republic of China

ML, 0000-0001-9765-3400; JZ, 0000-0002-6075-2373

Dilute magnetic semiconductors (DMSs), such as (In, Mn)As and (Ga, Mn)As prototypes, are limited to III–V semiconductors with Curie temperatures ($T_c$) far from room temperature, thereby hindering their wide application. Here, one kind of DMS based on perovskite niobates is reported. BaM$_x$Nb$_{(1-x)}$O$_{3-\delta}$ ($M$ = Fe, Co) powders are prepared by the composite-hydroxide-mediated method. The addition of $M$ elements endows BaM$_x$Nb$_{(1-x)}$O$_{3-\delta}$ with local ferromagnetism. The tetragonal BaCo$_x$Nb$_{(1-x)}$O$_{3-\delta}$ nanocrystals can be obtained by Co doping, which shows strong saturation magnetization ($M_{sat}$) of 2.22 emu g$^{-1}$, a remnant magnetization ($M_r$) of 0.084 emu g$^{-1}$ and a small coercive field ($H_c$) of 167.02 Oe at room temperature. The *ab initio* calculations indicate that Co doping could lead to a 64% local spin polarization at the Fermi level ($E_F$) with net spin DOS of 0.89 electrons eV$^{-1}$, this result shows the possibility of maintaining strong ferromagnetism at room temperature. In addition, the trade-off effect between the defect band absorption and ferromagnetic properties of BaM$_x$Nb$_{(1-x)}$O$_{3-\delta}$ is verified experimentally and theoretically.

## 1. Introduction

Since the discovery of ferromagnetism in Mn-doped InAs in 1992 [1], dilute magnetic semiconductors (DMSs) with doped transition metal elements have been increasingly studied due to their fascinating

**ROYAL SOCIETY OF CHEMISTRY**

multifunctional spintronic properties [2–4]. However, the solubility of doped transition metal atoms in a matrix remains a technical challenge, limiting these materials to the preparation of epitaxial films with metastable phase structures. Furthermore, elevating the $T_c$ of DMSs to room temperature is a long-standing request for their industrial application, and the complex carrier doping strategy can increase the $T_c$ of traditional DMSs to 100–180 K [5,6]. However, reports show that developing DMSs with transition metal oxides, such as $Zn_{1-x}Mn_xO_2$ and $Ti_{1-x}Co_xO2$, can extend the critical temperature above 300 K while maintaining a relatively low $M_{sat}$ ferromagnetism value of approximately $10^{-2}$ emu g$^{-1}$ [7,8].

Efforts to develop the room temperature DMSs in the form of powders or bulk ceramics have been carried out on Mn-doped ZnO; however, the formation of Mn clusters under high processing temperature ($T > 700°C$) would lead to suppression or disappearance in ferromagnetism [7]. Hence, the exploration of methods with lower synthesis temperature is of great importance to the industry application of DMSs in the bulk device and, extend the room temperature DMSs to other oxides system.

Perovskite niobates, with the composition of $ANbO_3$ (A = Li, K, Na, Ag), are widely used in lead-free piezoelectric and nonlinear optical devices due to their excellent ferroelectricity and nonlinear optical properties [9–14]. Previous theoretical and experimental work has proven that local ferromagnetism can be obtained in transition-metal-doped perovskite niobates [15–17]. Specifically, as a prototype of an A-site atom with an oxidation state of +2, $BaNbO_3$ has shown great potential as a room temperature DMS, showing that the introduction of oxygen defects, modulation of the electric field and Co doping with transition metals can be achieved; however, its $M_{sat}$ is limited to approximately $10^{-2}$ emu g$^{-1}$ [18–20].

# 2. Material and methods

## 2.1. Experimental procedure

### 2.1.1. Synthesis of *M*-doped BaNbO$_{3-\delta}$

*M*-doped BaNbO$_{3-\delta}$ nanocrystals were prepared by the composite-hydroxide-mediated method. In a typical synthesis, 3.8819 g of NaOH, 5.1181 g of KOH, 0.2660 g of $Nb_2O_5$ and 0.4880 g of $BaCl_2 \cdot 2H_2O$ were weighed and mixed, and then 0.2705 g of $FeCl_3 \cdot 6H_2O$, or 0.2379 g of $CoCl_2 \cdot 6H_2O$ was added into, respectively. The mixture was stirred and put in a 40 ml Teflon beaker. The Teflon beaker was placed in a preheated furnace at 195°C for 24 h. Then the Teflon beaker was taken out for natural cooling to room temperature. The reaction mixture was washed and filtered by distilled water and alcohol alternately three times. The filtered *M*-doped BaNbO$_{3-\delta}$ powder was dried at 60°C for 4 h.

### 2.1.2. Sample characterization

XRD was performed on a PANalytical Empyrean instrument outfitted with a PIXcel 2D detector operating at 40 kV per 40 mA, using Cu–Kα radiation ($\lambda = 1.5405$ Å). A Quanta 200FEG field emission SEM with EDS attachment was used for SEM analysis. XPS data were collected by ESCALAB 250Xi photoelectron spectrometer, which is produced by ThermoFisher company; the gun source was Al–Kα radiation. The HRTEM and SAED experiment was using a Tecnai G2 F30 transmission electron microscope. A Lake Shore 7404 vibrating sample magnetometer, with an external magnetic field sweeping from −15 000 to +15 000 Oe, was employed to evaluate the ferromagnetism of *M*-doped BaNbO$_{3-\delta}$ nanocrystals.

## 2.2. Theoretical calculation

DFT calculation in this study was performed with the Cambridge Serial Total Energy Package (CASTEP). A 2 × 2 × 2 supercell crystal model of pure $BaNbO_3$ was established first. Then one of the Nb atoms in the crystal was replaced by *M* elements, accompanied by one O vacancy in the oxygen octahedral cage. Localized density approximation (LDA) was employed for geometry optimization and property calculations with CA-PZ exchange-correlation functional. The plane-wave cut-off energy was set to 400 eV, and the *k*-point sampling grid was 4 × 4 × 4.

# 3. Results

Here, we report DMSs based on *M*-doped BaNbO$_{3-\delta}$, in which strong magnetism can be obtained at room temperature. The *M*-doped BaNbO$_{3-\delta}$ nanocrystals are prepared by the composite-hydroxide-

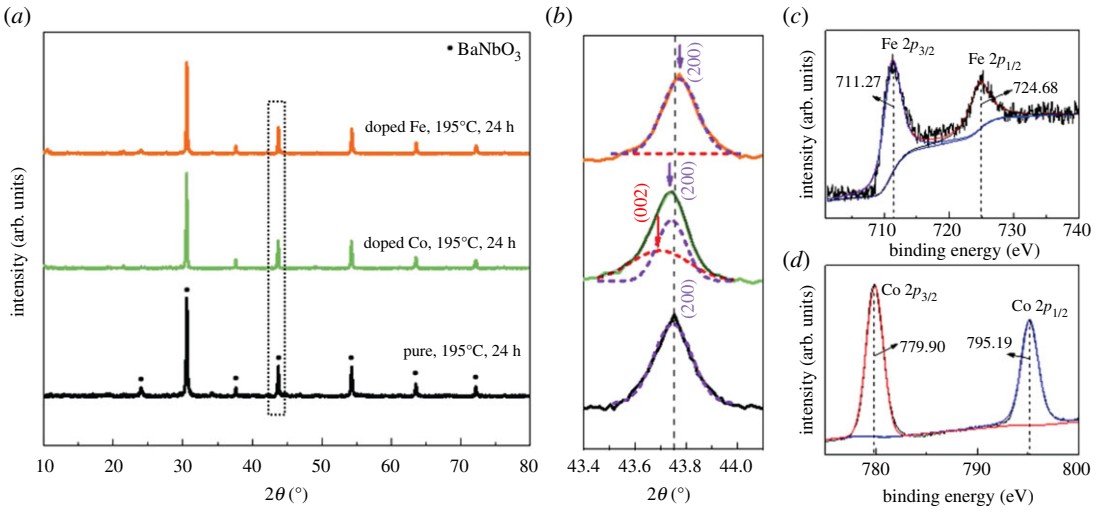

**Figure 1.** Crystal structure and elemental analysis of $BaM_xNb_{(1-x)}O_{3-\delta}$. (a) XRD patterns of the $BaM_xNb_{(1-x)}O_{3-\delta}$ nanocrystals. (b) Magnification of the (200) peaks in (a). (c,d) Elemental valence of the Fe and Co atoms in the $BaM_xNb_{(1-x)}O_{3-\delta}$ nanocrystals, respectively.

mediated (CHM) method. The as-synthesized pure $BaNbO_{3-\delta}$ has a cubic lattice with $a = b = c = 4.135$ Å (figure 1a,b) and belongs to the P-m3m space group. This lattice parameter is larger than the theoretical value of 4.080 Å. Additionally, this lattice expansion is beneficial for accommodating more dopants. It can be proven by the XRD results with $M$ doping that no second phase can be observed, as shown in figure 1a. The addition of Fe leads to a slight blue shift of diffraction peaks, indicating a decrease of the lattice parameter. While a red shift of diffraction peaks can be observed with the addition of Co, indicating an increase in lattice parameter, as shown in figure 1b. It is in accordance with the ionic radius of $Nb^{4+}$ (68 pm), $Fe^{3+}$ (64.5 pm) and $Co^{2+}$ (74.5 pm). In addition, the emergence of a shoulder diffraction peak at 43.6° (figure 1b) with Co doping indicates a phase change from the cubic $BaNbO_3$ to tetragonal $BaCo_xNb_{(1-x)}O_{3-\delta}$. The change in phase structure of $BaCo_xNb_{(1-x)}O_{3-\delta}$ powders indicates more Co atoms have been incorporated in the $BaNbO_3$ matrix, implying better ferromagnetic properties than $BaFe_xNb_{(1-x)}O_{3-\delta}$. Figure 1c,d shows the XPS results of $BaM_xNb_{(1-x)}O_{3-\delta}$ nanocrystal, and the oxidation states of Fe and Co are +3 and +2, respectively, combining with the EDS mapping measurements (electronic supplementary material, figure S1), corroborating that the dopants are successfully doped in the corresponding compounds. Furthermore, the existence of oxygen defects in the as-synthesized samples by CHM methods is proven by the blueshift in the split binding energy of the O_1s states (electronic supplementary material, figure S2). The oxygen defects are another origin for the emergence of room temperature ferromagnetism in $BaM_xNb_{(1-x)}O_{3-\delta}$ [18].

To obtain further insight into the doped crystal lattices, the high-resolution transmission electron microscopy (HRTEM) and selection area electron diffraction transmission electron microscopy (SAED TEM) are used to characterize the crystalline structure of as-grown $BaM_xNb_{(1-x)}O_{3-\delta}$ nanocrystals. As shown in figure 2a–c, the $BaFe_xNb_{(1-x)}O_{3-\delta}$ nanocrystal has identical lattice spacings of orthorhombic (101)/(100) and (200)/(002) facets, proving that the Fe-doped $BaNbO_{3-\delta}$ maintains the cubic phase. However, the addition of Co increases the facet of (101) to 2.939 Å and, changes the $BaCo_xNb_{(1-x)}O_{3-\delta}$ lattice to a tetragonal phase (figure 2e). It has a lattice spacing difference of 0.043 Å between (002) and (200) facets with an increased angle of approximately 91° (figure 2f), which matches well with the emergence of (002) peak in figure 1b.

# 4. Discussion

## 4.1. Room temperature ferromagnetism characterization

To evaluate the ferromagnetism of $BaM_xNb_{(1-x)}O_{3-\delta}$ nanocrystal, the field-dependent magnetization ($M$) was measured. Figure 3 shows the $M$ versus applied magnetic field ($H$) curve with field sweeping from −15 000 to +15 000 Oe. As shown in figure 3a, the addition of $M$ increases the ferromagnetism of $BaMxNb_{(1-x)}O_{3-\delta}$ nanocrystal, especially for tetragonal $BaCo_xNb_{(1-x)}O_{3-\delta}$, showing that an $M_{sat}$ of 2.22 emu g$^{-1}$ can be

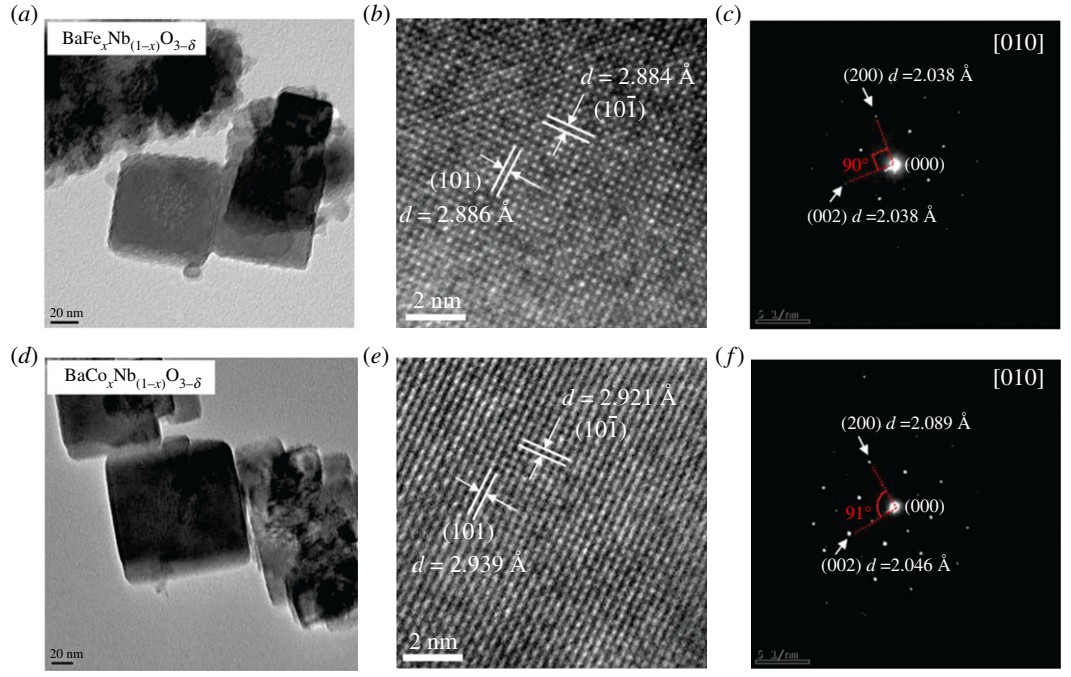

**Figure 2.** TEM characterizations of BaM$_x$Nb$_{(1−x)}$O$_{3−\delta}$ nanocrystals. (a,d) TEM images of Fe-, Co-doped the BaNbO$_{3−\delta}$ nanocrystals, respectively; (b,e), HRTEM images of Fe-, Co-doped the BaNbO$_{3−\delta}$ crystals, respectively; (c,f) corresponding SAED TEM images of samples.

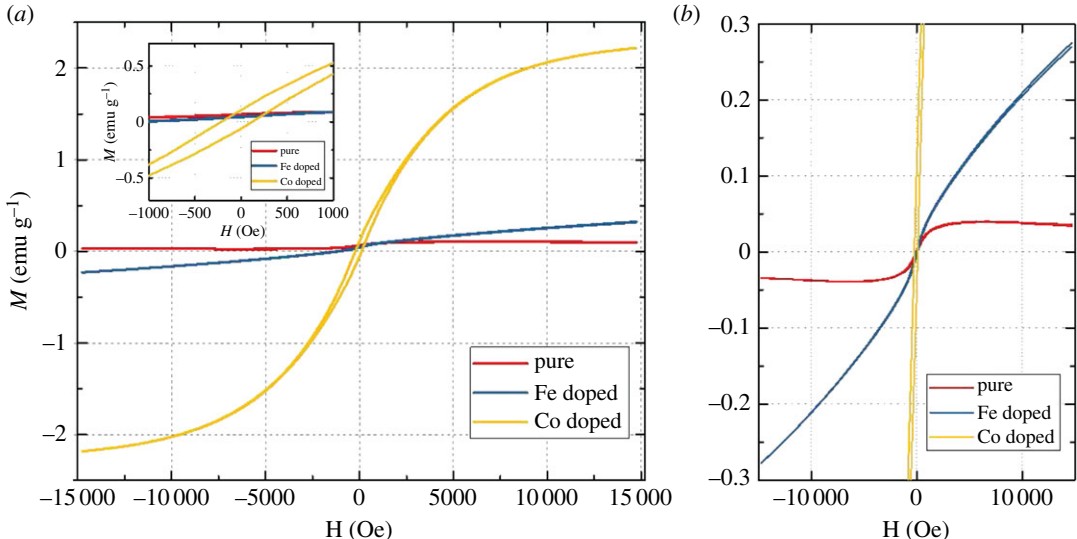

**Figure 3.** Magnetic properties of BaM$_x$Nb$_{(1−x)}$O$_{3−\delta}$. (a) Magnetic hysteresis curve $M(H)$, as measured in an external field $H$ of up to 15 kOe. The inset in the upper left corner shows the $M_r$ and $H_c$ of BaCo$_x$Nb$_{(1−x)}$O$_{3−\delta}$. (b) Magnified image shows the details of the $M(H)$ curves of pure BaNbO$_{3−\delta}$ and BaFe$_x$Nb$_{(1−x)}$O$_{3−\delta}$.

obtained at a driving field of approximately ±15 000 Oe, which is two orders of magnitude higher than that of pure BaNbO$_{3−\delta}$ (figure 3b). In addition, the BaFe$_x$Nb$_{(1−x)}$O$_{3−\delta}$ samples have an $M_r$ of 0.084 emu g$^{-1}$ and an $H_c$ of 167.02 Oe. Although the addition of Co and Fe increases the magnetization of the samples, their hysteresis loops present an 'S' shape, indicating a long-range magnetic order [21].

## 4.2. Mechanism of room temperature ferromagnetism

To understand the physical mechanism of strong ferromagnetism maintained at room temperature, we investigated the electronic structure of the prepared materials based on *ab initio* methods. The 2 × 2 × 2 supercell crystal models were built with $M$ elements replacing one of the Nb atoms, accompanied by

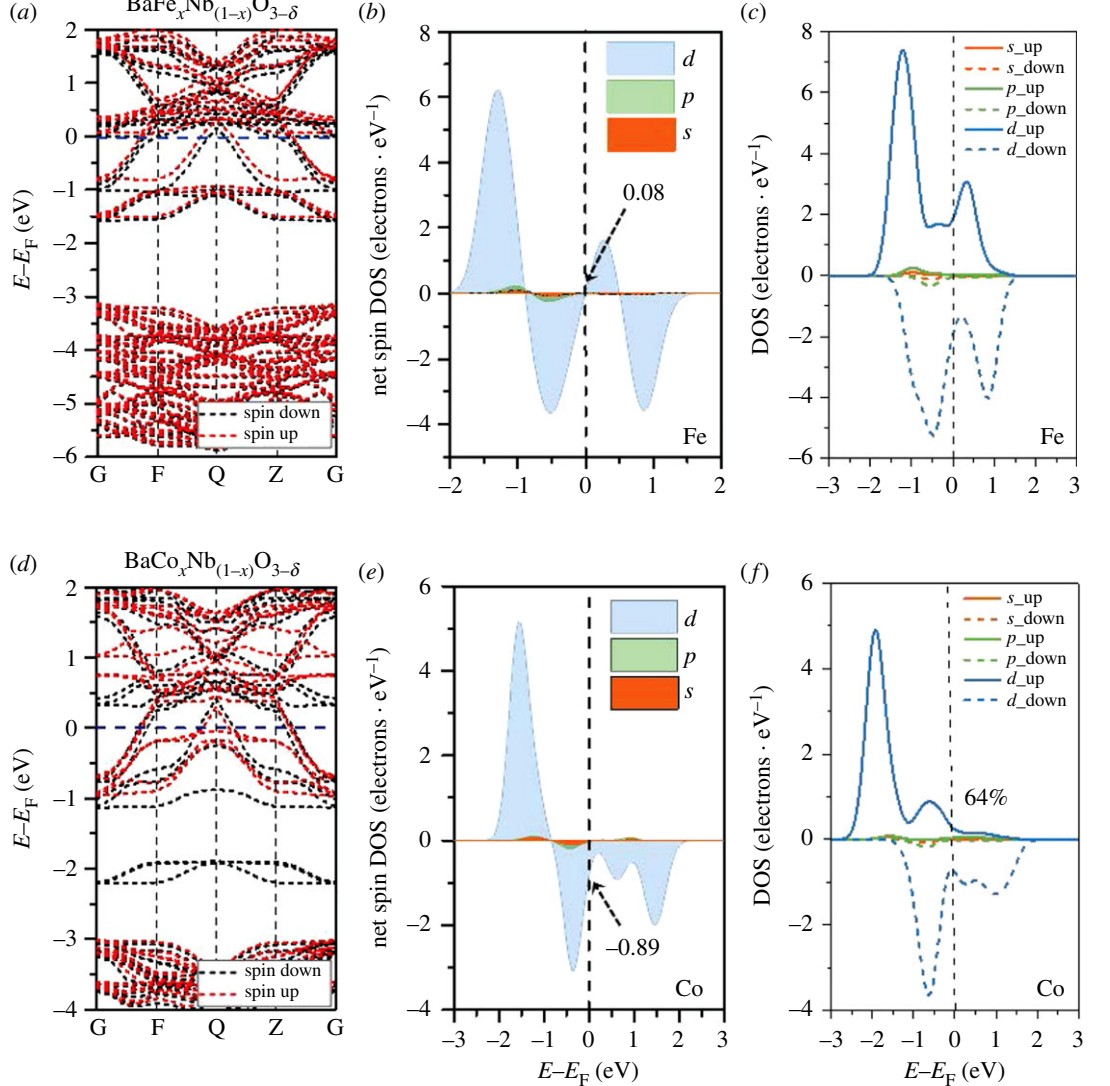

**Figure 4.** Calculated band structures and DOS of the doped atoms in the BaNbO$_{3-\delta}$ matrix. (a,d) Calculated band structures of Fe-, Co-doped BaNbO$_{3-\delta}$. (b,e) Calculated net spin PDOS of the Fe and Co atoms in the BaNbO$_{3-\delta}$ matrix, respectively. (c,f) Calculated spin-resolved PDOS of the Fe and Co atoms in the BaNbO$_{3-\delta}$ matrix, respectively.

one O vacancy in the oxygen octahedral cage, as shown in electronic supplementary material, figure S3. The calculated spin-resolved energy dispersion curves are shown in figure 4a,d. The carrier injection induced by doping transition metal elements and the existence of O defects lowers the conduction band minimum (CBM) below $E_F$, presenting metallic n-type conduction. A small correlating spin splitting can be observed in the band structure of the BaFe$_x$Nb$_{(1-x)}$O$_{3-\delta}$ model (figure 4a), while the addition of Co leads to strong correlating spin splitting of more than approximately 1 eV (figure 4d); this result means that Co doping will promote stronger ferromagnetism. The doped $M$ atom plays a dominant role in the spin polarization (see the supplemented partial density of states (DOS) of different atoms of BaM$_x$Nb$_{(1-x)}$O$_{3-\delta}$ in electronic supplementary material, figure S4), thereby presenting a local magnetic order. The calculated local net spin DOS of doped atoms at the $E_F$ of BaCo$_x$Nb$_{(1-x)}$O$_{3-\delta}$ is more than one order of magnitude higher than BaFe$_x$Nb$_{(1-x)}$O$_{3-\delta}$, of which the net spin DOS has the maximum absolute value of approximately 0.89 electrons eV$^{-1}$ (figure 4b,e). Thus, the predicted evolutionary trend remains in resolved DOS of doped atoms provides more details. Due to the weak correlating spin splitting of Fe, the major spin states almost overlap with the minor spin states, leading to a small spin polarization (figure 4c). The Co-doped BaNbO$_{3-\delta}$ shows the large spin splitting of approximately 1.26 eV, showing 64% spin polarization (figure 4f), which will contribute to maintaining strong ferromagnetism at room temperature.

Figure 5a shows the predicted optical absorption coefficient ($\eta$) of BaM$_x$Nb$_{(1-x)}$O$_{3-\delta}$ based on DFT calculations. The absorption induced by the transition between occupied O_2p orbitals and defect

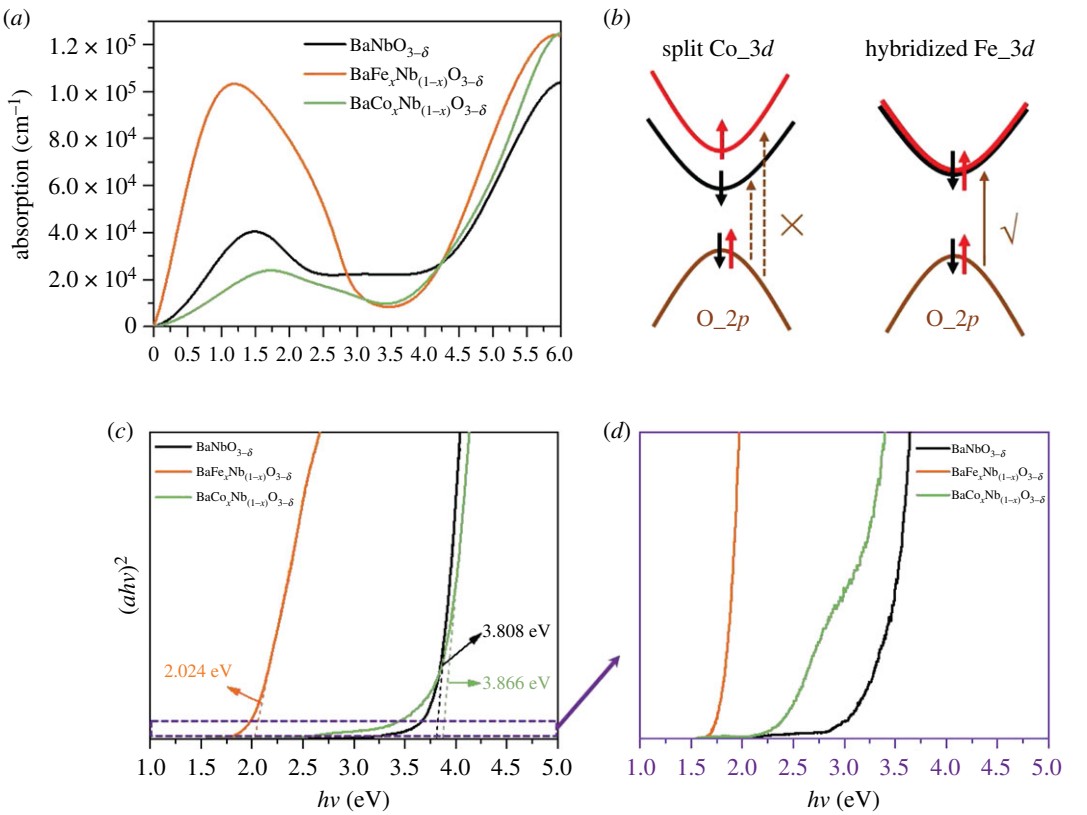

**Figure 5.** Absorption properties of $BaM_xNb_{(1-x)}O_{3-\delta}$. (a) Calculated absorption coefficients of pure $BaNbO_{3-\delta}$, $BaFe_xNb_{(1-x)}O_{3-\delta}$ and $BaCo_xNb_{(1-x)}O_{3-\delta}$. (b) Schematic showing the mechanism of the trade-off effect between the magnetic and absorption properties in $BaM_xNb_{(1-x)}O_{3-\delta}$. (c) UV–vis absorption spectra of the $BaM_xNb_{(1-x)}O_{3-\delta}$ nanocrystals. (d) The magnified region showing the defect band absorptions of the $BaM_xNb_{(1-x)}O_{3-\delta}$ nanocrystals.

bands can be observed in all of the models due to doping transition metal elements and the existence of O defects. Co doping shows a small contribution to the defect band absorption ($h\nu < 3.5$ eV), which is six times smaller than the major absorption between the O_2p and Nb_4d orbitals ($h\nu > 3.5$ eV). However, $BaFe_xNb_{(1-x)}O_{3-\delta}$ has a strong defect band absorption with a maximum $\eta$ higher than $10^5$, which is comparable to the absorption between the O_2p and Nb_4d orbitals. The large differences in the defect band absorption between Co and Fe doping are ascribed to the strong local spin splitting of Co in the $BaNbO_{3-\delta}$ matrix, while the Fe in the $BaNbO_{3-\delta}$ matrix shows weak local spin splitting with strong hybridization of the opposite spin states near $E_F$. As shown in the schematic of figure 5b, only states that have the same spin momentum are allowed, and the transition from degenerate O_2p orbitals to split Co_3d orbitals is forbidden. The above theoretical predictions are proven by the UV–vis absorption measurement results (spectra captured in a range of 1.55–6.22 eV). Figure 5c shows the absorption spectra of the prepared nanocrystal of $BaM_xNb_{(1-x)}O_{3-\delta}$. Co-doped $BaNbO_{3-\delta}$ have dominant absorption band edges of 3.866, which show absorption properties similar to those of pure $BaNbO_{3-\delta}$. However, Fe-doped $BaNbO_{3-\delta}$ has absorption band edges located at 2.024 eV, indicating strong defect band transition absorption. The weak defect band absorptions of $BaCo_xNb_{(1-x)}O_{3-\delta}$ and pure $BaNbO_{3-\delta}$ can be observed in figure 5d, and the results match well with the DFT calculations. The ferromagnetic performance and defect band absorption properties present a complementary relationship in $BaM_xNb_{(1-x)}O_{3-\delta}$ DMSs, of which strong ferromagnetism corresponds to a relatively weak defect band absorption. The weak defect band absorption further verifies that the orbits near the Fermi level of $BaCo_xNb_{(1-x)}O_{3-\delta}$ should have stronger Zeeman-type spin-polarization splitting.

# 5. Conclusion

Transition metal-doped $BaNbO_{3-\delta}$ nanocrystals are prepared by CHM methods. The as-prepared $BaCo_xNb_{(1-x)}O_{3-\delta}$ nanocrystals present better ferromagnetic properties than $BaFe_xNb_{(1-x)}O_{3-\delta}$.

The strong ferromagnetism is obtained at room temperature with an $M_{sat}$ of 2.22 emu g$^{-1}$, $M_r$ of 0.084 emu g$^{-1}$ and $H_c$ of 167.02 Oe, which is two orders of magnitude higher than that of pure BaNbO$_{3-\delta}$. The origin of the different ferromagnetic performances between Fe- and Co-doped BaNbO$_{3-\delta}$ nanocrystals is demonstrated by DFT calculations. Theoretical results show that the local net spin DOS of Fe and Co in the BaNbO$_{3-\delta}$ matrix are 0.08 and 0.89 electrons eV$^{-1}$, respectively. Moreover, the strong spin polarization (64%) and high binding energy difference of the opposite spin states (approx. 1.26 eV) will contribute to maintaining the large $M_{sat}$ of BaCo$_x$Nb$_{(1-x)}$O$_{3-\delta}$ at room temperature. The defective band transition properties of BaM$_x$Nb$_{(1-x)}$O$_{3-\delta}$ show a complementary relationship with their ferromagnetic performances since the band transition is forbidden between orbitals with different spin states, and this behaviour is corroborated by DFT calculations and absorption measurements. The tetragonal BaCo$_x$Nb$_{(1-x)}$O$_{3-\delta}$ within the morphology of bulk specimens provides a potential route for designing multifunctional spintronic devices that work at room temperature.

Data accessibility. Our data are deposited at the Dryad Digital Repository: https://doi.org/10.5061/dryad.f4qrfj6wd [22].

Authors' contributions. Y.Z., F.Z. and J.Z. designed and guided this study. Y.Z. wrote the main manuscript. Q.H. performed the calculation and data analyses. X.L. performed the TEM analysis. Y.L., Z.L., M.L. and T.H. helped with data analyses. Y.H. gave constructive suggestions during the calculation. All authors have given approval to the final version of the manuscript.

Competing interests. The authors declare no competing interests.

Funding. This work is supported by China Postdoctoral Science Foundation (grant no. 2019M651281).

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
