## [Peer Review File · Royal Society Open Science]

Review History

RSOS-210121.R0 (Original submission)

Review form: Reviewer 1

Is the manuscript scientifically sound in its present form?

Yes

Are the interpretations and conclusions justified by the results?

Yes

Is the language acceptable?

Yes

Do you have any ethical concerns with this paper?

No

Have you any concerns about statistical analyses in this paper?

No

Recommendation?

Accept with minor revision (please list in comments)

Comments to the Author(s)

The article entitled "Increased room temperature ferromagnetism in Co-doped tetrahedral perovskite niobates" by Zhou et al. is an interesting study can be accepted with minor revision. Authors shown the effect of metal doping (Fe and Co) in Barium niobates leads to ferromagnetism (in case of Co doping). Authors given the experimental and theoretical data for their results. Author should address the following comments:

1. XRD for all three cases (fig 1 a) should superimposed on each other and should check is there any significant shift in the peaks for higher theta.
2. Author should calculate crystallite size for all cases from XRD and also lattice strain from XRD.
3. It is not clear to see the shoulder peak in figure 1b at 43.6 degrees. Even for Fe-doped case i can see the shoulder peak. It will be better put XRD on log scale and include in manuscript which may help to see shoulder peak clearly.

Review form: Reviewer 2**Is the manuscript scientifically sound in its present form?**

Yes

Are the interpretations and conclusions justified by the results?

No

Is the language acceptable?

Yes

Do you have any ethical concerns with this paper?

No

Have you any concerns about statistical analyses in this paper?

No

Recommendation?

Major revision is needed (please make suggestions in comments)

Comments to the Author(s)

The paper is talking about Increased room temperature ferromagnetism in co-doped tetrahedral perovskite niobates. Although the authors did a good study, the discussion is not comparing the findings well. The authors are just giving the results what they obtained. Therefore, it would be better, if the discussion is elaborated a bit with some supporting evidence.

In addition, the peak in XRD corresponding to (002) plane is not clear and it cannot be claimed as indicated.

The tetragonal $\text{BaCo}_x\text{Nb}_{1-x}\text{O}_3-\delta$ within the morphology of bulk specimens seems to be a better candidate for designing multifunctional spintronic devices at room temperature. How this device is better than the existing devices. Are there any comparison studies done. How the device perform, if the temperature changes?

Authors have wealth of data, and those should be correlated well with their claims.

Introduction part should be further enhanced.

The novelty of this work should be indicated.

Kindly check the language throughout.

Decision letter (RSOS-210121.R0)

Dear Professor Zhu:

Title: Increased room temperature ferromagnetism in Co-doped tetrahedral perovskite niobates
Manuscript ID: RSOS-210121

Thank you for submitting the above manuscript to Royal Society Open Science. On behalf of the Editors and the Royal Society of Chemistry, I am pleased to inform you that your manuscript will be accepted for publication in Royal Society Open Science subject to minor revision in accordance with the referee suggestions. Please find the reviewers' comments at the end of this email.

The reviewers and handling editors have recommended publication, but also suggest some minor revisions to your manuscript. Therefore, I invite you to respond to the comments and revise your manuscript.

Because the schedule for publication is very tight, it is a condition of publication that you submit the revised version of your manuscript before 21-Jul-2021. Please note that the revision deadline will expire at 00.00am on this date. If you do not think you will be able to meet this date please let me know immediately.

- 1) A text file of the manuscript (tex, txt, rtf, docx or doc), references, tables (including captions) and figure captions. Do not upload a PDF as your "Main Document".
- 2) A separate electronic file of each figure (EPS or print-quality PDF preferred (either format should be produced directly from original creation package), or original software format)

- 3) Included a 100 word media summary of your paper when requested at submission. Please ensure you have entered correct contact details (email, institution and telephone) in your user account
- 4) Included the raw data to support the claims made in your paper. You can either include your data as electronic supplementary material or upload to a repository and include the relevant doi within your manuscript
- 5) All supplementary materials accompanying an accepted article will be treated as in their final form. Note that the Royal Society will neither edit nor typeset supplementary material and it will be hosted as provided. Please ensure that the supplementary material includes the paper details where possible (authors, article title, journal name).

Kind regards,
Dr Laura Smith
Publishing Editor, Journals

On behalf of the Subject Editor Professor Anthony Stace and the Associate Editor Dr Dattatray Late.

RSC Associate Editor:
Comments to the Author:
Accept with minor revisions

RSC Subject Editor:
Comments to the Author:
(There are no comments.)

Reviewer comments to Author:

Reviewer: 1

Comments to the Author(s)

The article entitled "Increased room temperature ferromagnetism in Co-doped tetrahedral perovskite niobates" by Zhou et al. is an interesting study can be accepted with minor revision.

Authors shown the effect of metal doping (Fe and Co) in Barium niobates leads to ferromagnetism (in case of Co doping). Authors given the experimental and theoretical data for their results. Author should address the following comments:

1. XRD for all three cases (fig 1 a) should superimposed on each other and should check is there any significant shift in the peaks for higher theta.
2. Author should calculate crystallite size for all cases from XRD and also lattice strain from XRD.
3. It is not clear to see the shoulder peak in figure 1b at 43.6 degrees. Even for Fe-doped case i can see the shoulder peak. It will be better put XRD on log scale and include in manuscript which may help to see shoulder peak clearly.

Reviewer: 2

Comments to the Author(s)

The paper is talking about Increased room temperature ferromagnetism in co-doped tetrahedral perovskite niobates. Although the authors did a good study, the discussion is not comparing the findings well. The authors are just giving the results what they obtained. Therefore, it would be better, if the discussion is elaborated a bit with some supporting evidence.

In addition, the peak in XRD corresponding to (002) plane is not clear and it cannot be claimed as indicated.

The tetragonal $\text{BaCo}_x\text{Nb}_{1-x}\text{O}_3-\delta$ within the morphology of bulk specimens seems to be a better candidate for designing multifunctional spintronic devices at room temperature. How this device is better than the existing devices. Are there any comparison studies done. How the device perform, if the temperature changes?

Authors have wealth of data, and those should be correlated well with their claims.

Introduction part should be further enhanced.

The novelty of this work should be indicated.

Kindly check the language throughout.

Author's Response to Decision Letter for (RSOS-210121.R0)

See Appendix A.

RSOS-210121.R1 (Revision)

Review form: Reviewer 1

Is the manuscript scientifically sound in its present form?

Yes

Are the interpretations and conclusions justified by the results?

Yes

Is the language acceptable?

Yes

Do you have any ethical concerns with this paper?

Yes

Have you any concerns about statistical analyses in this paper?

No

Recommendation?

Accept as is

Comments to the Author(s)

All comments has been addressed well. Article can be accepted.

Review form: Reviewer 2

Is the manuscript scientifically sound in its present form?

Yes

Are the interpretations and conclusions justified by the results?

Yes

Is the language acceptable?

Yes

Do you have any ethical concerns with this paper?

No

Have you any concerns about statistical analyses in this paper?

No

Recommendation?

Accept as is

Comments to the Author(s)

The comments raised are addressed carefully. The paper is now in a good shape and can be considered for publication after a thorough check of English.

Decision letter (RSOS-210121.R1)

Dear Professor Zhu:

Title: Increased room temperature ferromagnetism in Co-doped tetrahedral perovskite niobates

Manuscript ID: RSOS-210121.R1

It is a pleasure to accept your manuscript in its current form for publication in Royal Society Open Science. The chemistry content of Royal Society Open Science is published in collaboration with the Royal Society of Chemistry.

Yours sincerely,
Dr Ellis Wilde
Publishing Editor, Journals

On behalf of the Subject Editor Professor Anthony Stace and the Associate Editor Dr Dattatray Late.

RSC Associate Editor
Comments to the Author:
Accept as is

RSC Subject Editor
Comments to the Author:
(There are no comments.)

Reviewer(s)' Comments to Author:
Reviewer: 2

Comments to the Author(s)
The comments raised are addressed carefully. The paper is now in a good shape and can be considered for publication after a thorough check of English.

Reviewer: 1
Comments to the Author(s)
All comments has been addressed well. Article can be accepted.

Appendix A

Manuscript ID: RSOS-210121

TITLE: **Increased room temperature ferromagnetism in Co-doped tetrahedral perovskite niobates**

Dear Dr. Laura Smith:

Thank you very much for handling our manuscript and sending us the reviewer's reports on our manuscript (Paper Ref. No.: RSOS-210121). We also thank the reviewers for reading our manuscript and providing valuable comments. We have revised the original manuscript and supplementary according to your and the reviewers' suggestions. The revised manuscript and supplementary are attached for your approval. The point-to-point responses to the referees' comments are listed in the last part of this letter.

According to your comments, we made corresponding changes as follows:

1. A text file of the manuscript (tex, txt, rtf, docx or doc), references, tables (including captions) and figure captions. Do not upload a PDF as your "Main Document".

Reply: Thanks for the comment. We have uploaded the new test file as request.

2. A separate electronic file of each figure (EPS or print-quality PDF preferred (either format should be produced directly from original creation package), or original software format)

Reply: Thanks for the comment. We have uploaded figure files as request.

3. Included a 100 word media summary of your paper when requested at submission. Please ensure you have entered correct contact details (email, institution and telephone) in your user account

Reply: Thanks for the comment. We have prepared media summary as “This research reports a kind of DMS based on perovskite niobates. Local ferromagnetism has been found in \$\text{BaM}_x\text{Nb}_{(1-x)}\text{O}_{3-\delta}\$ (\$M = \text{Fe, Co}\$ ) powders prepared by CHM method. The tetragonal \$\text{BaCo}_x\text{Nb}_{(1-x)}\text{O}_{3-\delta}\$ shows strong saturation magnetization (\$M_{\text{sat}}\$ ) of 2.22emu/g, a remnant magnetization (\$M_r\$ ) of 0.084emu/g, and a small coercive field (\$H_c\$ ) of 167.02Oe at room temperature. DFT calculations indicate that Co doping could lead to a 64% local spin polarization at the Fermi level (\$E_F\$ ) with net spin DOS of 0.89 electrons·eV⁻¹. The trade-off effect between the defect band absorption and ferromagnetic properties of \$\text{BaM}_x\text{Nb}_{(1-x)}\text{O}_{3-\delta}\$ is verified experimentally and theoretically.” Also, we have checked our user account, and the contact details are correct.

4. Included the raw data to support the claims made in your paper. You can either include your data as electronic supplementary material or upload to a repository and include the relevant doi within your manuscript

Reply: Thanks for the comment. We have uploaded the raw data to Dryad (<https://doi.org/>

[10.5061/dryad.f4qrfj6wd](https://doi.org/10.5061/dryad.f4qrfj6wd)) as request.

5. All supplementary materials accompanying an accepted article will be treated as in their final form. Note that the Royal Society will neither edit nor typeset supplementary material and it will be hosted as provided. Please ensure that the supplementary material includes the paper details where possible (authors, article title, journal name).

Reply: Thanks for the comment. We have checked the supplementary file to make sure it meets the requirements.

We look forward to receiving your decision soon.

With best regards,

Jingchuan Zhu

Department of Materials Science and Engineering

Harbin institute of technology

Harbin, China, 150001

Email: fgms@hit.edu.cn

The point-to-point responses to the referees' comments

(For clarity, the referees' comments are cited in *italics*)

Reviewer #1:

1. XRD for all three cases (fig 1 a) should be superimposed on each other and should check if there is any significant shift in the peaks for higher theta.

Reply: Thanks for the comment. In order to check the peak shift with elements doping, the dashed line is used to mark the peak shift (see revised Fig. 1b). The significant blue-shift can be observed with the addition of Fe, while the red-shift can be observed with the addition of Co. It is in accordance with the atom radius of Nb⁴⁺ (68 pm), Fe³⁺ (64.5 pm) and Co²⁺ (74.5 pm).

2. Author should calculate crystallite size for all cases from XRD and also lattice strain from XRD.

Reply: Thanks for the comment. The XRD in the work is used to prove the phase structures of as-grown samples. Although the crystallite size could be estimated by fitting the diffraction peaks, it could not be applied in our case. As shown in Figure R1, the crystallite sizes of Fe-doped, Co-doped, and pure BaNbO₃ are range from 100 nm to 500 nm. We used the SEM data to depict the crystallite sizes, which also can be observed in supplementary Fig. S1.

Figure R1. (a), (b), (c) SEM images of the $\text{BaNbO}_{3-\delta}$, $\text{BaFe}_x\text{Nb}_{(1-x)}\text{O}_{3-\delta}$ and $\text{BaCo}_x\text{Nb}_{(1-x)}\text{O}_{3-\delta}$ powders, respectively. (scale bars are 200 nm)

3. It is not clear to see the shoulder peak in figure 1b at 43.6 degrees. Even for Fe-doped case i can see the shoulder peak. It will be better put XRD on log scale and include in manuscript which may help to see shoulder peak clearly.

Reply: Thanks for the comment. To clearly show the shoulder peak of (002), we use gaussian function to fit the peak of (002) and (200), as shown in the revised Fig.1b. Only the Co-doped case could obtain the should peak of (002).

Reviewer #2:

1. The paper is talking about Increased room temperature ferromagnetism in co-doped tetrahedral perovskite niobates. Although the authors did a good study, the discussion is not comparing the findings well. The authors are just giving the results what they obtained. Therefore, it would be better, if the discussion is elaborated a bit with some supporting evidence.

Reply: Thanks for the comment. To demonstrate our claims clearly, we increased the discussion of the theoretical predictions and experimental data and, building the correlations between them. Such as “The addition of Fe leads to a slight blue shift of diffraction peaks, indicating a decrease of the lattice parameter. While a red shift of diffraction peaks can be observed with the addition of Co, indicating an increase in lattice parameter, as shown in Fig. 1b. It is in accordance with the ionic radius of Nb^{4+} (68pm), Fe^{3+} (64.5pm) and Co^{2+} (74.5pm).”, “The change in phase structure of $\text{BaCo}_x\text{Nb}_{(1-x)}\text{O}_{3-\delta}$ powders indicate more Co atoms have been incorporated in the BaNbO_3 matrix, implying better ferromagnetic properties than $\text{BaFe}_x\text{Nb}_{(1-x)}\text{O}_{3-\delta}$.”, “The oxygen defects is another origin for the emergence of room temperature ferromagnetism in $\text{BaM}_x\text{Nb}_{(1-x)}\text{O}_{3-\delta}$.^[18]”, “The weak defect band absorption further verifies that the orbits near Fermi lever of $\text{BaCo}_x\text{Nb}_{(1-x)}\text{O}_{3-\delta}$ should have stronger Zeeman-type spin polarization splitting.”.

2. The peak in XRD corresponding to (002) plane is not clear and it cannot be claimed as indicated.

Reply: Thanks for the comment. To clearly show the shoulder peak of (002), we use gaussian function to fit the peak of (002) and (200), as shown in the revised Fig.1b. Only the Co-doped case could obtain the should peak of (002).

3. The tetragonal $\text{BaCo}_x\text{Nb}_{(1-x)}\text{O}_{3-\delta}$ within the morphology of bulk specimens seems to be a better candidate for designing multifunctional spintronic devices at room temperature. How this device is

better than the existing devices. Are there any comparison studies done. How the device perform, if the temperature changes?

Reply: Thanks for the comment. The method for preparation of DMSs is dominated by the epitaxial method currently, the challenge of increasing the solubility of doped transition metal atoms and the metastable phase structure limit their wide application, as shown in the discussion in the introduction part. In addition, To demonstrate it clearly, we added the discussion in the introduction part as “Efforts to develop the room temperature DMSs in the form of powders or bulk ceramics have been carried out on Mn-doped ZnO, however, the formation of Mn clusters under high processing temperature ($T > 700^\circ\text{C}$) would lead to suppression or disappearance in ferromagnetism.^[7] Hence, the exploration of methods with lower synthesis temperature is of great importance to the industry application of DMSs in the bulk device and, extend the room temperature DMSs to other oxides system.”.

4. Authors have wealth of data, and those should be correlated well with their claims.

Reply: Thanks for the comment. We further added the discussion and enhanced the correlation of the data between each part. Such as “The change in phase structure of $\text{BaCo}_x\text{Nb}_{(1-x)}\text{O}_{3-\delta}$ powders indicates more Co atoms have been incorporated in the BaNbO_3 matrix, implying better ferromagnetic properties than $\text{BaFe}_x\text{Nb}_{(1-x)}\text{O}_{3-\delta}$.”, “The oxygen defects is another origin for the emergence of room temperature ferromagnetism in $\text{BaM}_x\text{Nb}_{(1-x)}\text{O}_{3-\delta}$.^[18]”, “The weak defect band absorption further verifies that the orbits near Fermi level of $\text{BaCo}_x\text{Nb}_{(1-x)}\text{O}_{3-\delta}$ should have stronger Zeeman-type spin polarization splitting.”.

5. Introduction part should be further enhanced.

Reply: Thanks for the comment. We have enhanced the discussion of the development of bulk DMSs in the introduction part.

6. The novelty of this work should be indicated.

Reply: Thanks for the comment. The novelty of this work is that we report a kind of DMSs based on perovskite niobates within the form of nanoparticles. Tetragonal phase can be obtained in Co-doped $\text{BaNbO}_{3-\delta}$. It shows strong ferromagnetism with (M_{sat}) of 2.22 emu/g, remnant magnetization (M_r) of 0.084 emu/g, and coercive field (H_c) of 167.02 Oe at room temperature. The obtained M_{sat} at room temperature is two orders of magnitude higher than that of pure $\text{BaNbO}_{3-\delta}$, which can be seen in the conclusion.

7. Kindly check the language throughout.

Reply: Thanks for the comment. We have double-checked the grammar errors and polished the language.